# IoT-Based Cotton Plant Pest Detection and Smart-Response System

**Saeed Azfar** [1], **Adnan Nadeem** [2,*], **Kamran Ahsan** [1], **Amir Mehmood** [3], **Hani Almoamari** [2] **and Saad Said Alqahtany** [2]

1 Department of Computer Science, Federal Urdu University of Arts, Science and Technology, Karachi 75300, Pakistan
2 Faculty of Computer and Information Systems, Islamic University of Madinah, Medina 42351, Saudi Arabia
3 Department of Computer Science and IT, Sir Syed University of Engineering and Technology, Karachi 75300, Pakistan
* Correspondence: adnan.nadeem@iu.edu.sa

**Abstract:** IoT technology and drones are indeed a step towards modernization. Everything from field monitoring to pest identification is being conducted through these technologies. In this paper, we consider the issue of smart pest detection and management of cotton plants which is an important crop for an agricultural country. We proposed an IoT framework to detect insects through motion detection sensors and then receive an automatic response using drones based targeted spray. In our proposed method, we also explored the use of drones to improve field surveillance and then proposed a predictive algorithm for a pest detection response system using a decision-making theory. To validate the working behavior of our framework, we have included the simulation results of the tested scenarios in the cup-carbon IoT simulator. The purpose of our work is to modernize pest management so that farmers can not only attain higher profits but can also increase the quantity and quality of their crops.

**Keywords:** IoT-based pest management; sensors; cotton pest detection; smart response system

## 1. Introduction

With the increase in population, the demand for food and crops is also increasing. However, due to water scarcity and climate change and the area of fertile land is decreasing. At the same time, pests and diseases remain the enemies of the crop and its quality [1,2].

Cotton is one of the most important crops, whether we talk about its commercial importance or mankind's need for it. Today, a large part of every country's economy depends on its crops, and cotton is indeed a very widely grown crop. According to the International Cotton Advisory Committee's report, Pakistan is the fifth largest producer of cotton after India, China, the USA, and Brazil [1,3]. However, cotton is also the favorite crop of insects, so wherever it is planted, there is a high risk of damage by insects. Diseases also affect its growth and production as well as its quality. Therefore, it is our social and state responsibility to make effective plans for the control of cotton pests and diseases so that we can attain good quality cotton in great quantities [4–6].

In recent times we have seen that the Internet of Things (IoT) has become a very fast-spreading technology. The Internet of Things has quickly made inroads into various areas of life, such as health, communications, manufacturing, and energy, as well as agriculture. IoT has seen improvements in all these areas, and the quality of its products and services is increasing. [7,8] We also know that the future of technology is very bright in agriculture. The features of IT sensors, LORA communication modules, and drones, which we can use in farming or agriculture, could not be more impressive in any other field. Due to these features, the combination of IoT and farming is indeed becoming very popular.

Pest identification and its prevention is a field of work in which IoT is indeed very useful. However, many farmers are still reluctant to adopt this technology despite its usefulness. There are many reasons for this; for example, Figure 1 below shows some reasons. These barriers cannot be categorized as either big or small because they are all equally important. However, one of the most important apprehensions in adopting this technology is that it will add trouble and hassle to traditional farmers dealing with the new technology. Therefore, we believe that there is a need to study further the use of IoT-based architectures and prototypes for pest management which will contribute to both enhancing food quality and its production.

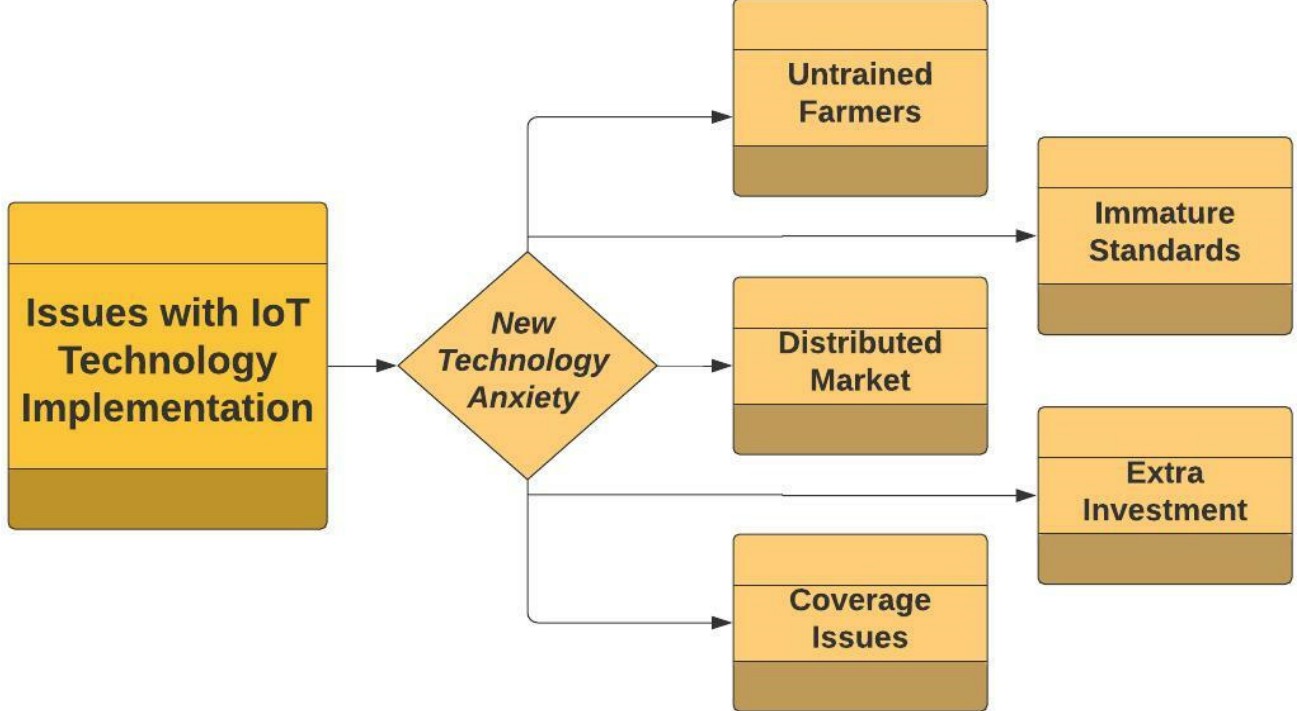

**Figure 1.** Major issues in IoT implementation in Agriculture.

Due to rapid enhancement in technology, there is a dire need for a predictive system that will enable farmers to increase their crop field as well as improve their quality and profit. In this paper, we propose an IoT framework in which sensors and drones will work together to protect cotton crops from flying insects and cotton bollworms. It can be used for real-time monitoring of agricultural fields. In our previous work, we proposed a locally assembled sensor prototype to detect cotton bollworm pests and validated it through real-time implementation both in the field and in the lab. In the study in this paper, we propose an IoT-based cotton plant pest detection and smart response system. This will be very useful for farmers in terms of minimizing losses as well as improving the quality of the crops.

The novelty of the proposed system is that it is a complete system. It covers the entire process of detecting cotton pests and response, i.e., destroying them through pesticides with the help of drones. We have also proposed an algorithm for the complete flight movement of drones based on decision-making theory. The proposed system is general and can be applied to other crop pests by selecting suitable sensors.

The rest of the paper is organized as follows. Section 2 discusses the related work of technologies used in smart agriculture and IoT-based models for crop pests in the literature. Our proposed system is described in Section 3 in detail, and the drone flight predictive algorithm is included as an appendix. Simulation and its results are presented in Section 4. The last section concludes our work and ideas for future work.

## 2. Related Work

This section consists of the existing literature in the field of agriculture. These studies comprised various AI technologies, IoT sensors, and flying drones.

### 2.1. Technologies Used in Agriculture

A very comprehensive survey has been presented which examines the benefits of the IoT as well as the challenges and benefies associated with this technology [9–11]. Fields and orchards are generally located far from the population, making it almost impossible for the farmer to look after them all the time, and the power supply is very limited due to the remoteness of the infrastructure. This makes it very difficult to apply the technologies, and the farmer is afraid to apply such things. As a result, even simpler and more energy-efficient technologies, such as the Internet of Things, have been finding it difficult to make their mark in agriculture.

Titiya et al. [12] have developed an expert system based on ontology to control the presence of cotton diseases and pests. In this system, the cotton farmer inputs his queries, and the system helps him. It has been proven that this system can answer very difficult questions and suggest solutions to all these difficult queries.

Xiao et al. [13] proposed a recurrent neural network-based (RNN) system in which pests and diseases are identified due to various weather factors. It includes a huge network that saves memory for a short time. The predictive results of this proposed network are very good, indicating many diseases and pests.

Li et al. [14] have also come up with a model for diagnosing diseases that is based on artificial intelligence. A Neural Network is used in this system, and they also have developed a web-based intelligent diagnostic system that will be used to control cotton diseases. The accuracy of the predictions of this system is more than 85%, and the time required is very short, i.e., 900 ms.

In another study [15], Gao et al. presented an agricultural framework that applied to wheat crops. Figure 2 shows the proposed IoT platform, in which they have taken pictures of different parts of the crop through the Internet of Things and drones and sent them to the cloud data center to identify the environment conducive to pests in the field due to climate change. In this framework, we see that as the crop grows, the sensors collect real-time data related to the crop. Due to the size of the fields, which are large and delivering energy to the fields is a difficult task, the researchers have installed a solar system to provide energy to the sensors, which provides constant power, and the drones flying down have taken pictures of the separate parts of the field, and the same images are uploaded to the cloud that can be processed to assess the field environment and crop with the help of spectrum analysis technology. Due to the flight mode, the flight time of the drones has been extended, and the changes in the crop due to pests or diseases are monitored in the data center with the help of pictures.

It has been observed that the probability of the presence of the disease is higher due to the temperature being between 14 °C and 16 °C, and when there is a lot of rain, the chance of spreading powdery mildew disease is reduced.

In another study [16], Vanegas et al. presented a methodology in which they used Unmanned Aerial Vehicle (UAV) integrated with advanced digital hyperspectral, multi-spectral, and RGB sensors. He proposed a method in which ground and aerial surveys were carried out by drones and sensors. They designed a predictive model that decides different levels of Phylloxera infestation based on airborne RGB, multi and hyperspectral imagery with ground-based data at two different periods.

In [17], a system for diagnosing cotton pests and diseases has been proposed. It is shown in Figure 1 of [17], a combination of two different technologies that combine case-based reasoning and physio-logic to design and implement a web-based system.

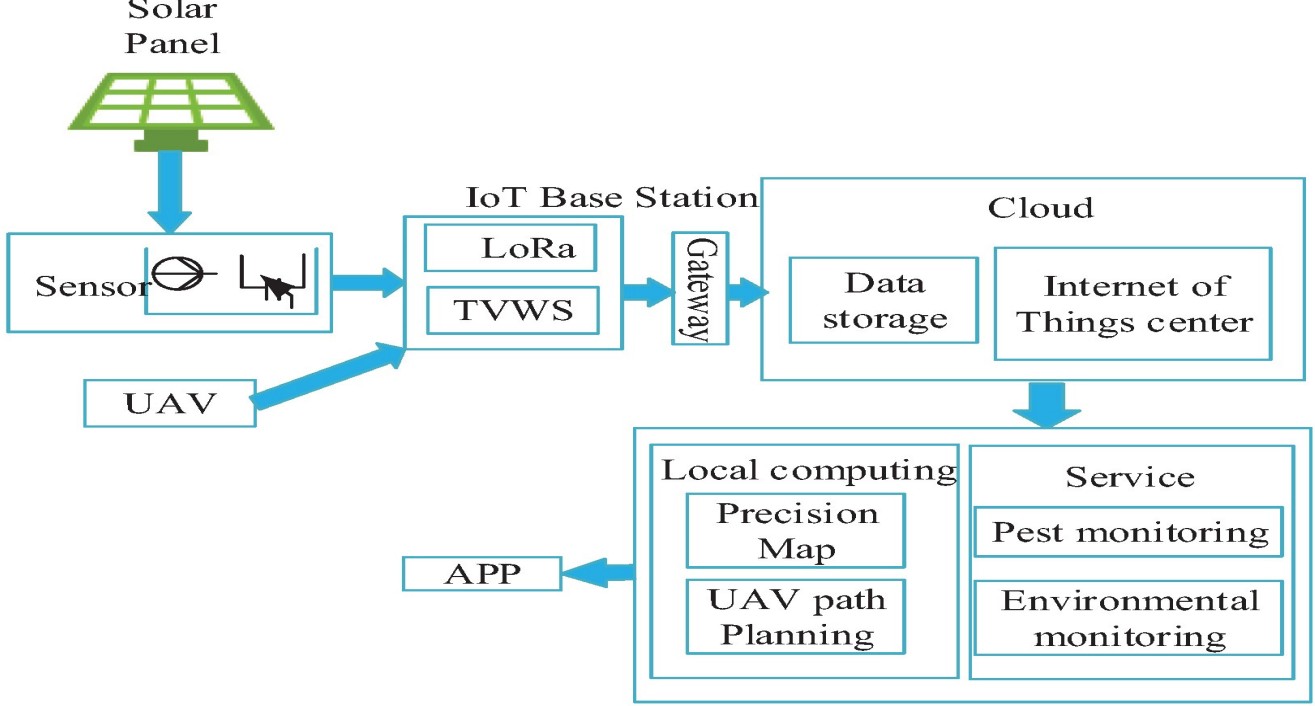

**Figure 2.** Overview of proposed Agriculture IoT platform [15].

In another study [18], Researchers have tried to detect Red Palm weevil and related diseases through a prototype. The weevil is a very destructive insect that is capable of destroying the entire palm crop. It is very difficult to see with the naked eye, and when it is visible, it is too late. Therefore, it is essential to use new technology to protect the crop from this pest. In this work, scientists have developed a prototype to prove their concept and planted it in the crop, and tried to monitor it from a distance around the clock so that the red palm weevil can be identified in time and, at the same time, it can be destroyed. It produced three kinds of performance parameters such as success, incomplete, and failure. The results were analyzed, and the success rate was over 90%.

In [19], the researchers have proposed an IoT-based system that, in conjunction with digital image processing, will detect various plant leaf diseases. In this system, some sensors monitor the values of various parameters of the environment, while many diseases can be detected by taking pictures with the help of digital cameras and processing digital images on them. This system will work on a somewhat limited scale as the camera cannot take pictures at night while the sensors and server machine also have an energy problem.

This research [20] has proposed an IoT framework that will monitor eight specific field parameters on a daily basis. A predictive model has also been developed that has been able to identify diseases and insects through machine learning.

Sharma R. P et al. [21] proposed a Fuzzy-Inference Crop Pest Prediction system. Various Rice and sugarcane/Millet pests and related disease were identified with the help of multiple weather parameters. Figure 2 of [21] shows the proposed layed architecture. The measured parameter includes temperature, humidity, atmospheric pressure, and rainfall. a Virtual Network Connection (VNC) is formed through the use of 5G-LTE connectivity. The proposed system predicts the occurrence of Rice pests using a fuzzy-logic-based mechanism.

In [22], the authors proposed a hardware design, a network of wireless sensors, and an actuator network, which is a great application in the field of agriculture. The researchers have proven that in the development of modern technology, they have not denied the importance of remote-monitoring technology drones, and if combined with

image processing technology, it can be a great tool for agriculture. With the help of this, the crabs and diseases can be seen from afar and, at the same time, take any steps to cure them.

In [23], Polo et al. have suggested a sensor network used in a large field that will provide coverage through nodes spread across the field. A mobile node in the form of a drone collects data whose job is to collect data. Once a video is made, it has to send it to the data center. The researchers claim that their research is very standardized and much cheaper than the existing technology in the market.

In Table 1, we have tried to give a quick overview of all the modern academic research on the subject.

**Table 1.** Existing IoT frameworks and Models for Pest and Disease detection.

| Research | Proposed System | Country | Crop | Pest or Disease | Contribution |
|---|---|---|---|---|---|
| Tatya et al. [12] | Ontology-based Expert System. | India | Cotton | Multiple Pests and Diseases. | They have suggested an Ontology-based expert system to answer related to pests and diseases using this system which uses Web-based technologies. |
| Xiao et al. [13] | Recurrent Neural Networks (RNN), a Class of Artificial Neural Network | China | Cotton | Multiple Pests and Diseases. | This research has aimed to predict pests and diseases using a long-short-term memory (LSTM) network. This system converts pest occurrence problems into time series problems and predicts the occurrence of pests and diseases. |
| Li et al. [14] | BP Neural network and web-based Expert system | China | Cotton | Multiple Diseases | Based on a BP-based neural network algorithm, a web technologies-based system has been designed in which a database of diseases has been developed, and decisions are made through the interference engine. The accuracy of the system is about 80–90%. |
| Gao, D. et al. [15] | Agriculture Framework based on IoT and UAV. | China | Wheat | Multiple diseases | Attempts have been made to create an agricultural framework that shows the relationship between weather components and diseases. The sensors of the proposed system run on solar energy and a new flight mode have also been used for drones. |
| Vanegas, F. et al. [16] | Predictive Model for insect detection | Australia | Grape | Grape Phylloxera | They have used a UAV integrated with advanced digital hyperspectral, multispectral, and RGB sensors. He has proposed a method in which ground and aerial surveys were carried out by drones and sensors by collecting data through various surveys. |
| Dong, Y. et al. [17] | Disease Diagnostic system. | China | Cotton | Multiple Diseases | They have designed and proposed a cotton disease diagnostic system (CDDS), which is based on case-based reasoning (CBR) and fuzzy logic. This system diagnoses diseases and proposes treatment. If we provide at least four symptoms as input to this system, it will give more than 90% accurate results. |

**Table 1.** *Cont.*

| Research | Proposed System | Country | Crop | Pest or Disease | Contribution |
|---|---|---|---|---|---|
| Koubaa, A. [18] | IoT Framework | Saudia Arabia | Palm/Dates | Red Palm Weevil | They have proposed a framework in which a smart palm monitoring prototype has been developed, and it monitors palms remotely using smart IoT-based agriculture sensors, which are capable of detecting red palm infestation in its early stages. It also uses web or mobile applications to interact with farms remotely. They have used an industrial-level IoT platform to interface between the user and system layers. |
| Apeksha, T. [19] | IoT and DIP-based Framework | India | Multiple Crops | Multiple Diseases | Researchers have proposed an IoT-based system that, in conjunction with digital image processing, detects various plant leaf diseases. In this system, some sensors monitor the values of various parameters of the environment, while many diseases can be detected by taking pictures with the help of digital cameras and processing digital images on them. |
| Materne, N. [20] | IoT Framework | Indonesia | Multiple | Multiple | This research has proposed an IoT framework that will monitor eight specific field parameters on a daily basis. A predictive model has also been developed that has been able to identify diseases and insects through machine learning. |
| Sharma, R. P. [21] | IoT Framework | India | Rice and Sugarcane | Multiple | WSN Monitoring Infrastructure-Driven Fuzzy-Logic-Based Crop Pest prediction proposed. In which pests and diseases are identified due to various weather factors. The measured parameter includes temperature, relative humidity, and rainfall. The acquired weather information is used as training data by the genetic algorithm (GA) to optimize the rule base of a fuzzy logic-based prediction system. |

In [24], the authors have proposed a system for early warning in a city that would use sensors to collect values for important environmental factors such as air pressure, temperature, rainfall values, and records by sensing the moisture. Then all of these factors, along with their values, will be uploaded to the cloud via a wireless network and a pre-existing model on the server that will examine the possibility of the presence of diseases using the Big Data Analysis method. Proposed system is early warning system for pests and diseases.

## 2.2. IoT in Agriculture

The rapid technological development in recent years in IoT technologies is going to play a vital role in many agriculture-related applications. It is just because of the services and capabilities offered by the internet of things. Most authors have reviewed that

most IoT architectures consist of four layers, the perception/sensors or device layer, the transport/network layer, which includes protocols, the processing or services layer, which offers processing and analysis of data and the application layer in last which includes monitoring and control applications [25–39]. These layers include all the main components of any IoT solution.

In a paper [25], they proposed a framework for *Phytophthora* early detection. It also maps out the future of how this technology will determine the future of agriculture and what will be the factors that will hinder it and prevent its spread in the future, and what things will challenge it in the future. These papers discuss the trends of IoT and also say which issues should be resolved very soon; otherwise, there will be no benefit in adopting this technology. At the same time, people's hopes for this technology and the limitations of this technology have been discussed. Figure 3 provides the key drivers of technology in smart agriculture. In these studies [25,26], researchers have made a very comprehensive review of how IoT devices have completely transformed farming similar to any other field.

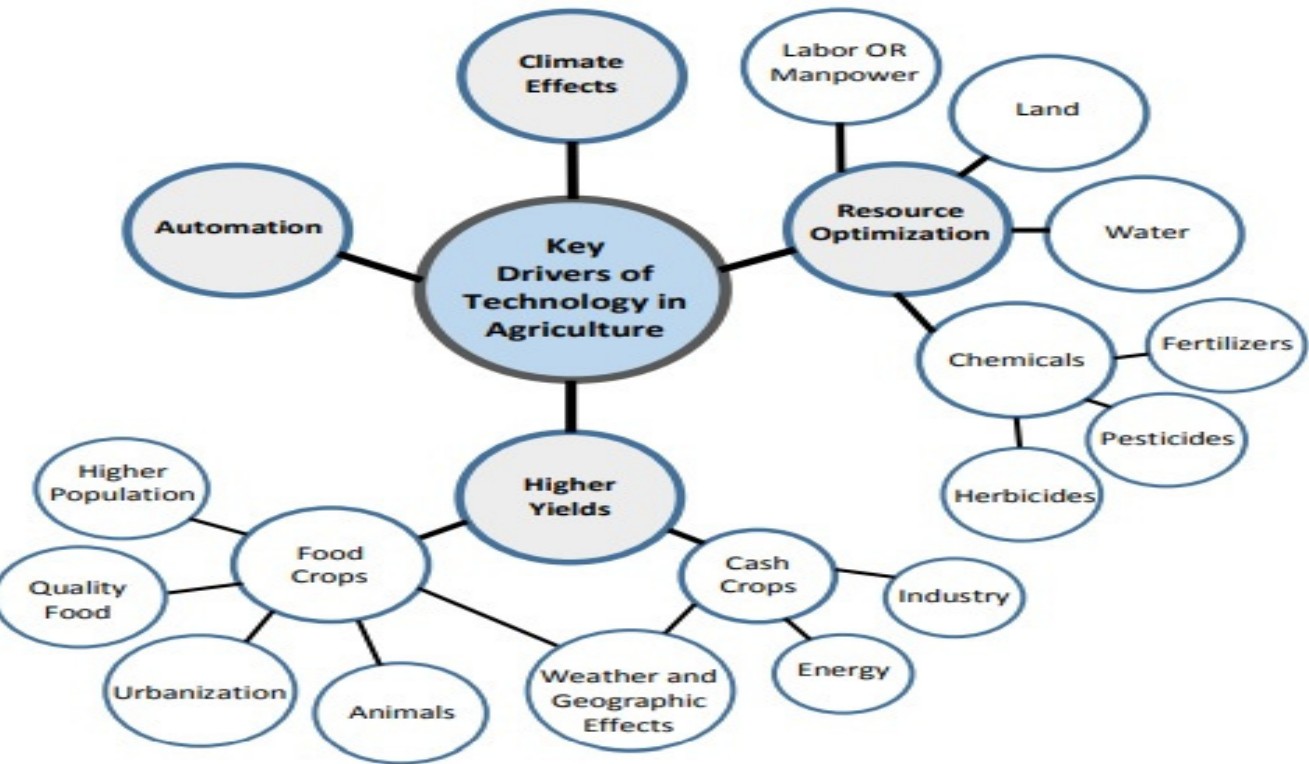

**Figure 3.** Key drivers of Technology in Agriculture [26].

We know that researching a new technology can be difficult and exciting. In [27], the authors have come up with the idea of using Light Detection and Ranging (LiDAR) sensors in drones to view the environment from above and survey the fields. Similarly, a 3-D imaging system has been proposed in [28] for agriculture applications.

A real-time system has been developed in [29] that has made videos of pests and diseases and has analyzed those using sensors and video cameras. It also discusses the new trends in agriculture, i.e., sensors, and tries to prevent pests and diseases by obtaining data with the help of these sensors. In this project, the farmer can view the data of his entire field sensors through a web application or mobile application. In this work, they have used an industrial-level user interface. After receiving data from sensors, researchers have used some signal processing tools, while some statistical tools have also been part of the work.

Forest fires are a natural calamity, and they have disasterous impacts on forests. An IoT-based system [30] has been proposed to control and monitor forest fires.

In [31], the authors have discussed IoT implementation in agriculture. It has covered a very comprehensive survey that has covered this research in different parts of the world where IoT has been used to identify crop insects or diseases. Combinations of different technologies have been discussed with IoT. Different types of frameworks and models are mentioned, and many hardware and sensors or prototypes are mentioned. A dataset paste24 is also mentioned, as well as a cloud-based smart farming system, while other technologies that have been used so far are also mentioned. While in [32], an IoT framework has been proposed to monitor different diseases and pests through weather parameters.

In a detailed IoT technology discussion related to agriculture available in [33], the researcher discussed several aspects of IoT technologies. Such as why we should use IoT in agriculture and associated challenges. Figure 4 shows a detailed portion of specialized sensors and their possible use in IoT-based smart agriculture. Related technologies, Drones and IoT-based tractors, and harvesting robots were also discussed, along with communication technologies used in IoT-based smart farming.

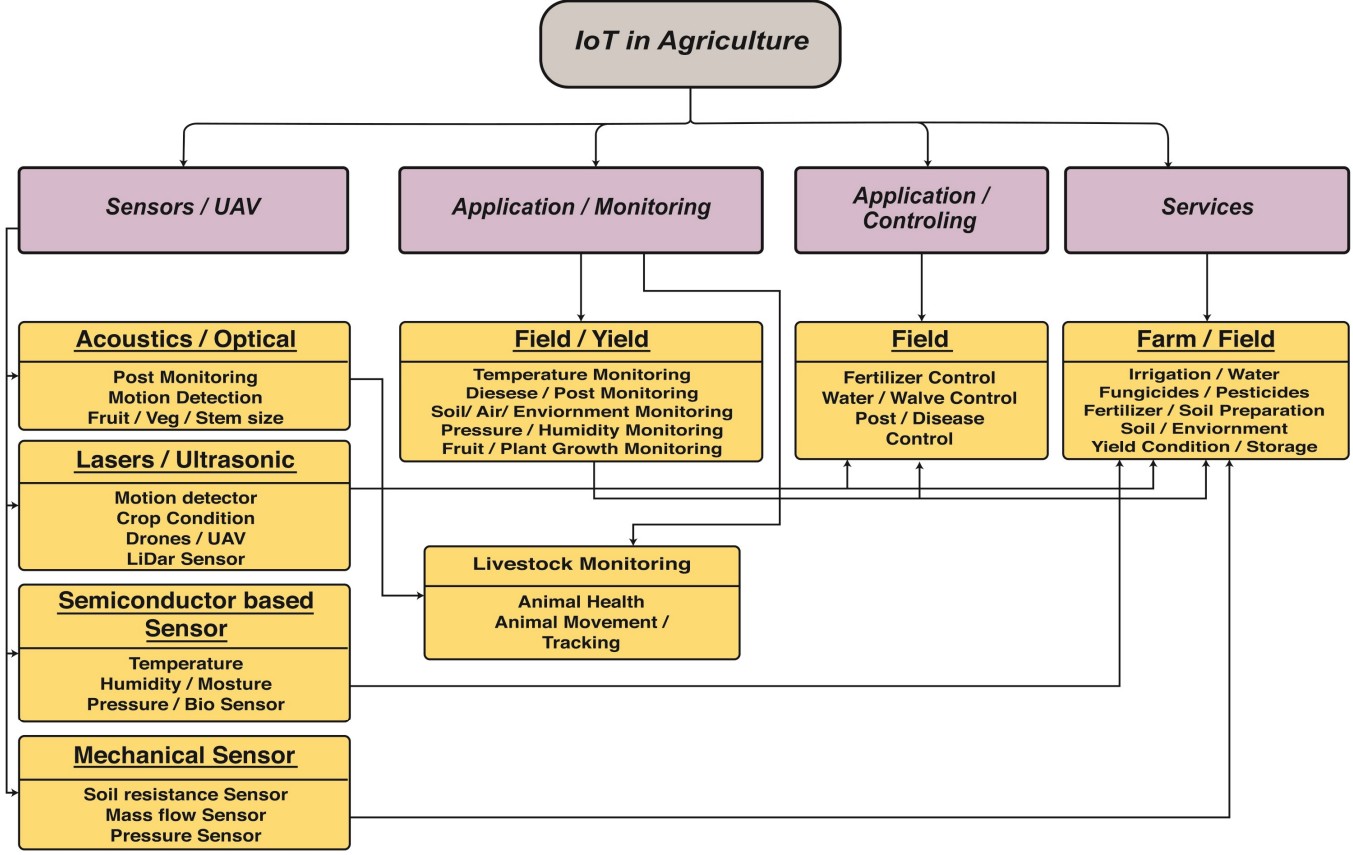

**Figure 4.** IoT-based Sensor' technologies and applications.

In [34], the authors have discussed selected aspects of IoT in agriculture, sensors, network technologies, and standard communication protocols. The authors have presented IoT based smart farming framework, which has five major components, including data acquisition, platforms, processing with visualization, and complete system management. Major company's products were also mentioned. Open issues and challenges were discussed, and IoT integration with the cloud was also discussed. Some surveys are available, but in [35], different types of surveys have been surveyed, and they have been categorized according to time. In the early days of the survey, communication protocols and communication technologies were discussed, while later, the architecture of IoT was explained, while the best applications were mentioned in the last round of surveys.

Another recent research study about pest and disease management using IoT is [36]; in this article, the authors have designed an IoT-based agriculture framework in which they have explained the relationship among pest, disease, and weather parameters. They have also explained the use of drones to attain different types of parameters, such as farmland images, etc. In this work, they have designed a sun tracker device to track sunlight, a new flight mode has been introduced to use wind force to prolong the flight time of UAV, and the image captured by Drones transmitted to the cloud for analysis and degree of damage inflicted by pests and diseases based on spectrum analysis technology. Last, they have deployed their designed framework somewhere in China in wheat crops.

In [37], the researchers proposed farming architecture to monitor different irrigation parameters of farms and detect animals using a Convolution Neural Network (CNN). They used three different IoT-based sensors for temperature, moisture, and motion detection. Temperature and moisture sensor for farm irrigation while motion detector for animal intrusion system. Two algorithms for farm irrigation and animal intrusion in the farm were proposed, and android based application for keeping farm owners updated. The authors claimed the system performed better in terms of accuracy.

A report [38] has been published in the recent past, which explores the impact of density and vegetation height on RSSI when two nodes are communicating with each other. They deployed IEEE 802.15.4 enabled nodes in Potato and Wheat crops to investigate path loss of 2.4 GHz Radio signals. Initial node deployment experiences disconnectivity and signal attenuation due to the density and height of wheat vegetation.

They proposed Non-Dominated Sorting Genetic Algorithm (NSGA-II) based optimized Node Deployment Strategy (NDS) for Wheat and Potato farming. Scientists monitor the significant difference between the NSGA II-NDS and existing NDS Strategy is that to monitor the Path Loss Coefficient for a crop to be monitored before deployment to decrease the possibility of a link dis-connectivity.

Sharma, R. P. et al. [39] proposed an IoT-Enabled IEEE 802.15.4 WSN Monitoring Infrastructure-Driven Fuzzy-Logic-Based Crop Pest Prediction. In this proposed architecture, pests and diseases are identified due to various weather factors. The measured parameter includes temperature, relative humidity, and rainfall. The acquired weather information is used as training data by the genetic algorithm (GA) to optimize the rule base of a fuzzy logic-based prediction system. Researchers deployed the low-power consumption IEEE 802.15.4 CC2538-2592 SoC microcontroller to establish a network of sensor nodes in Millet and Rice Crops. The proposed system predicts the occurrence of Rice pests using a fuzzy logic-based mechanism. The predictive results of this proposed network are very good, indicating many diseases and pests.

If we compare the use of IoT in agriculture [20,21,25,26,31–35,38,39], we will see that researchers have conducted great work and tried to improve agriclutural operations using various latest technologies. Some researchers are detecting the presence of pests based on weather parameters. Some are directly informed about pests through specific sensors.

The only difference between our research and the work of others is that we are proposing a complete system. The proposed system works with the identification of pests until their elimination from crops. Everything will be automated, and the response system will perform its operations automatically. Target is to minimize the use of pesticides and improve the quality of yield.

## 3. IoT-Based Smart Detection and Response System Architecture

In this section, we describe our proposed cotton crop pest detection and a smart response in detail. In our initial work, we developed a prototype system to detect cotton bollworm pests. Our proposed device consisted of Sharp's high-resolution motion-detecting sensors and a ZigBee communication module connected with an Arduino ATMega module, and a used Arduino editor and Sketch to program all these modules.

We tested this detection prototype in three phases. In the first stage, we tested in a controlled environment in the university research lab for 15 days, and the results were

very encouraging. Then, we deployed it in the cotton fields for several days, and again we observed a reasonably good detection rate. The sensors in the prototype send a detection packet containing data in Hexa decimal values, including its coordinates, to gateway devices, and it tells us the location of the sensor, and we would know in which part of the tested area has the presence of pests. It took us four months to complete this process. It was a small-scale test due to the unavailability of a large number of sensors.

To test large-scale implementation, we used a cup-carbon simulator [40] to test various scenarios. In the simulation, we program a device as a drone to test our smart response system after detecting pests. We proposed a drone movement prediction algorithm for targeted spray. In the future, we plan to implement the prosed pest detection and response system in large cotton fields, subject to the availability of sufficient funds.

### 3.1. Core Functionality of System

We are outlining our proposed system here, in which we have divided the system into certain sections for the convenience of those who understand and would implement this system in the fields. Our proposed detection-response system is shown in Figure 5.

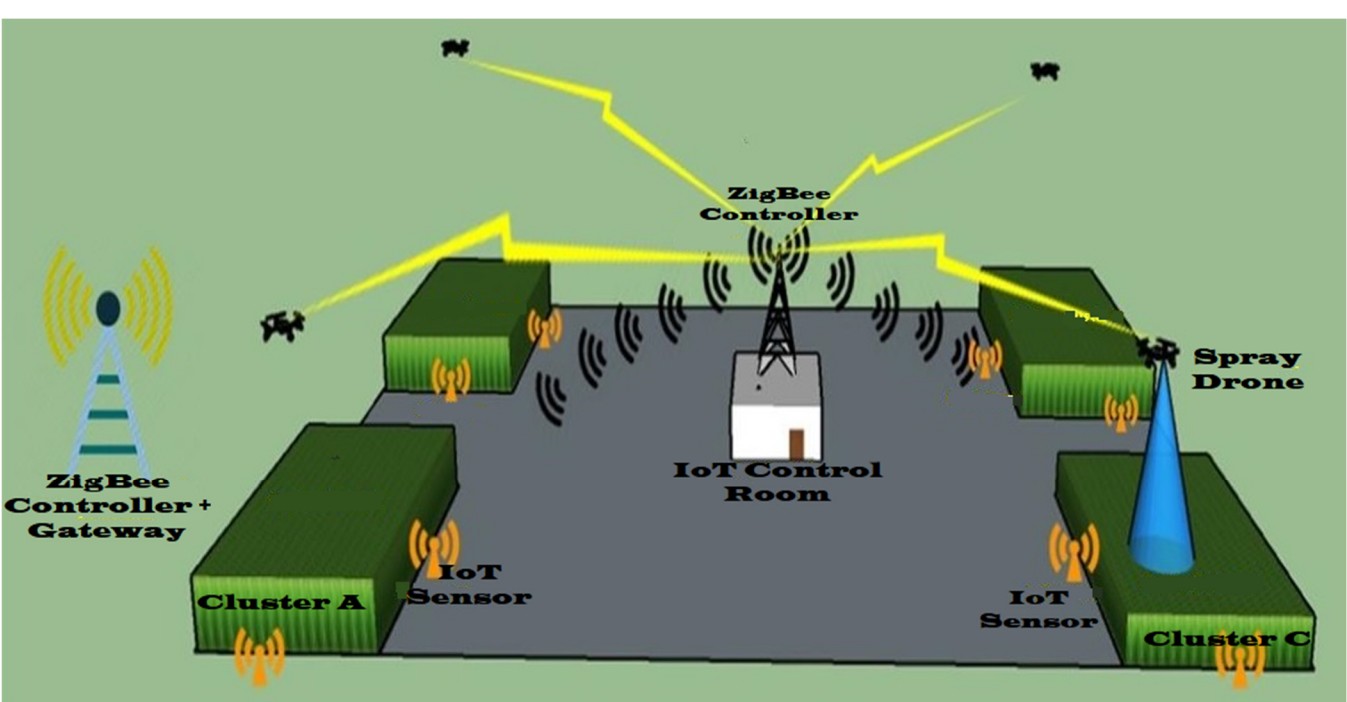

**Figure 5.** Pest Detection and Response system architecture.

Figure 6 shows the flow of the entire system, from the sensor to the base station and then to the drone, how the data and signals are moving and which devices are interfering. The core functions of our system include identifying the insect's presence near the crop, sending a detection packet through the LoRa communication module of its presence to the base station, and attempting to destroy it by targetted spray through a drone in response. Its framework and structure are shown in Figure 5, while Figure 6 shows the flow of signals throughout the system.

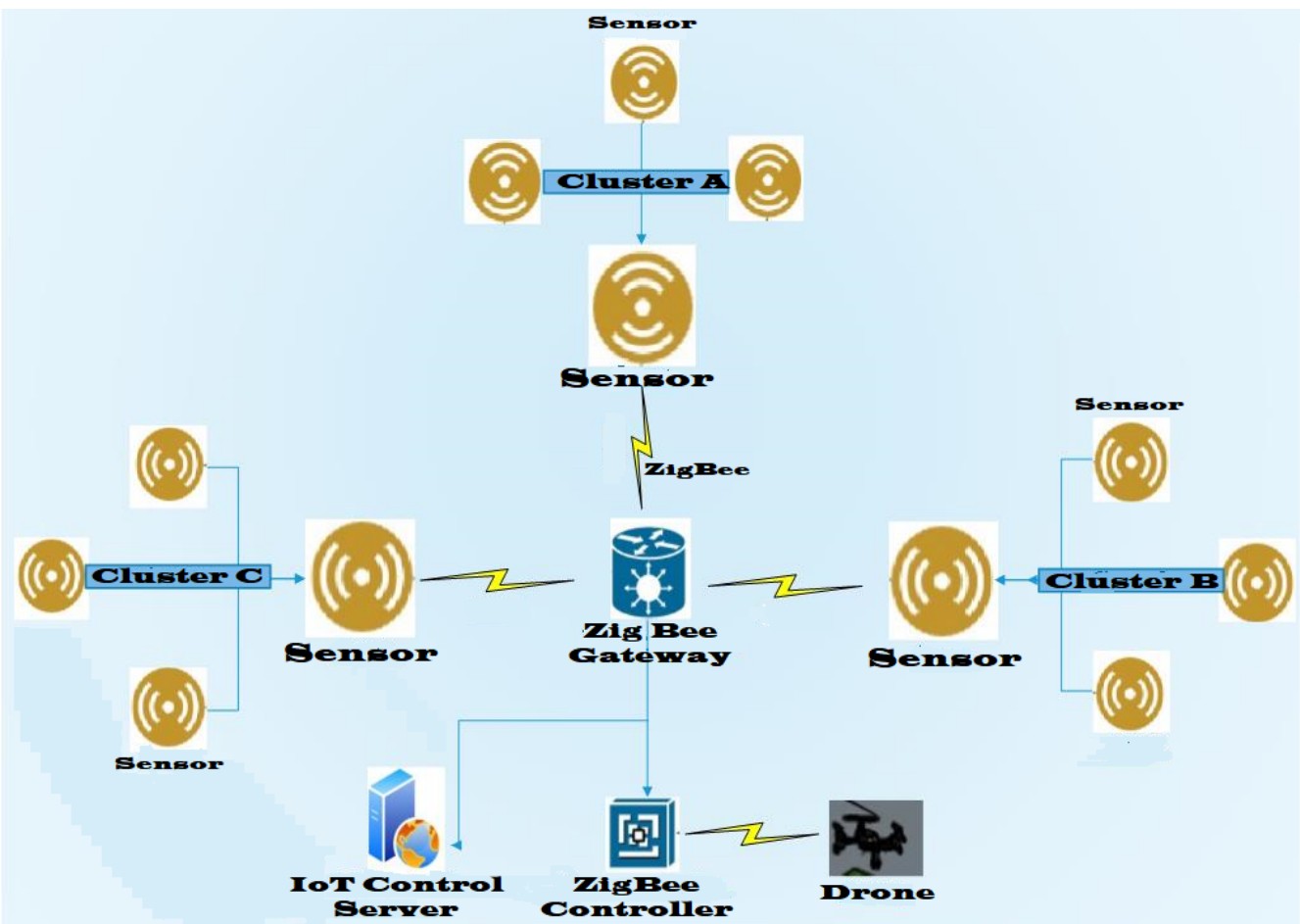

**Figure 6.** Detection-Response system flow diagram.

### 3.2. IoT-Based Cotton Pest Detection

Our proposed system consists of five tiers, as shown in Figure 7, in which the first tier is responsible for sensing the data indicative of insects while the second layer will analyze this data, and the third layer is based on this analytical data towards the sensor that sends the signal to the drone. The fourth layer will be drones that will go and spray the location from where the signal is received.

*The Field Tier:* This is one of the most basic layers in the cotton field. It will have sensors and communication modules. The responsibility of this layer is to detect the presence of insects through sensors, and as soon as the presence of an insect is detected, it will transmit the signal of its presence to the base station through the communication module, as shown in the detection Algorithm see Appendix A. (detection algorithm consists of a Serial transmitter and receiver while the sensor's pin represents a separate sensor. Sensor values are threshold value between one to four as per the manufacturer's specs, which means one represent high sensitivity while four shows low sensitivity. Distance defines as an object's distance with a sensor.)

Along with the sensor, the prototype is equipped with a ZigBee transmitter (ZED) that transmits signals, and we know that this ZigBee module covers a range of 10–100 m line of sight with 250 kbps data rates, while if we use LoRA instead, our coverage and data carrying capacity will increase. As shown in Figure 5, we have divided the whole field into four clusters, and with each cluster, we have installed sensors according to its area. Each sensor is capable of sending signals to the ZigBee gateway, which is located in the center of the field. We have shown a gateway in Figures 7 and 8, but we can install more than one gateway when needed.

**Figure 7.** Detection-Response Methodology Flow chart.

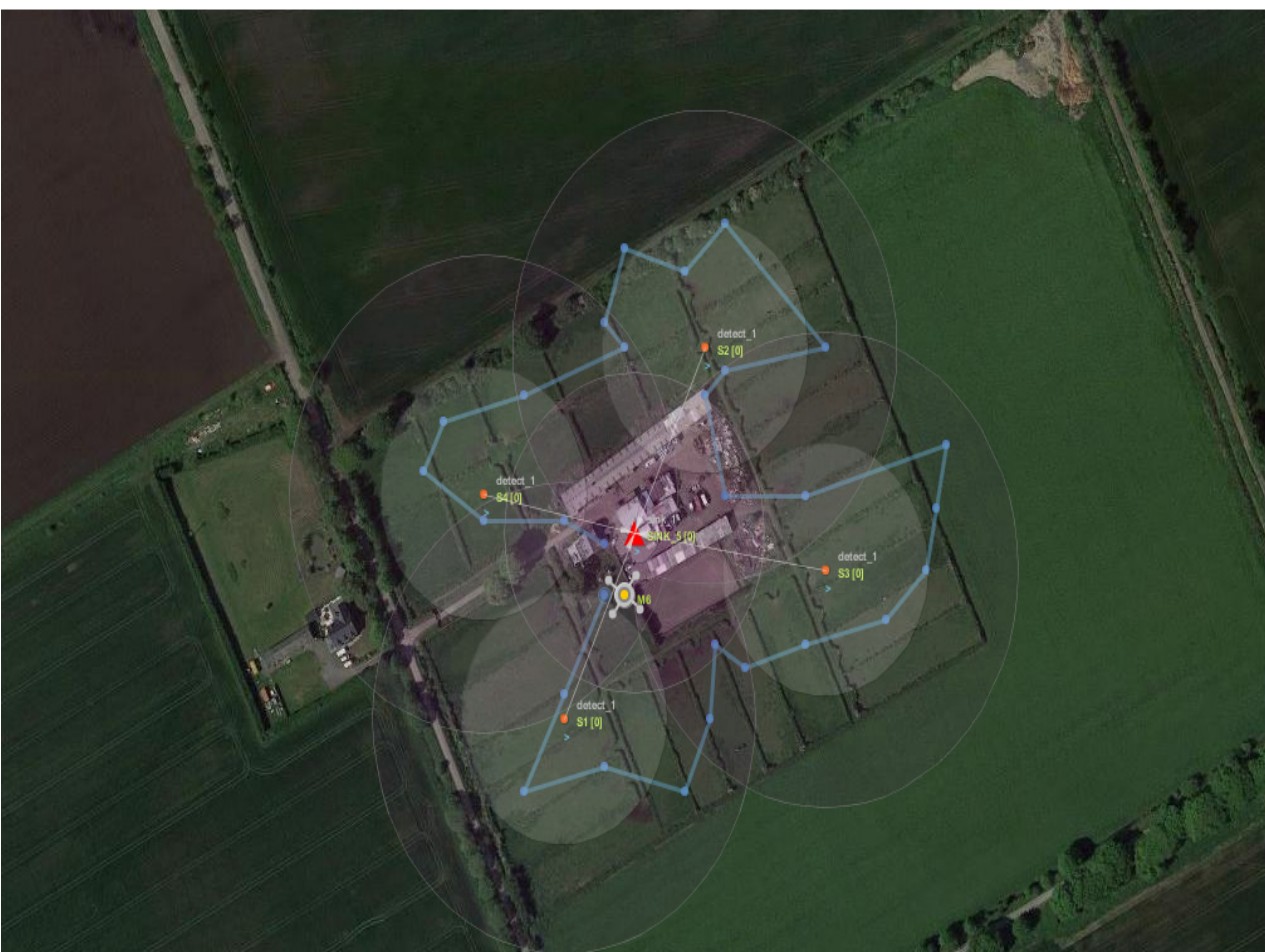

**Figure 8.** Cup Carbon Simulation field, which consists of Clusters.

- ***The Gateway Tier***: The function of this layer is to collect the data sent by the sensors and respond accordingly. When sensors send messages about the presence of insects, it decides where to send the drones. For example, if a message is received once from a sensor and repeated messages are received from a sensor. It will send the drone to the side where the messages were coming from again and again. Because we are using ZigBee and we have not yet extended the range of our architecture, the whole system can be controlled from one or two central ZigBee Gateways, but if we use LoRa Gateway, it will be more manageable, and we will have to adopt it because it has a longer range and more data carrying capacity than ZigBee.

- ***The Control Tier:*** This tier will respond to the detection data packet from the sensors.

  We perform the following analysis based on this data

I    First, in which cluster did more insects move in and what could be the cause?

II   Second, the crop quality of this cluster may be different from other clusters.

III  Third, more spray is required in clusters with a high number of insect detections.

IV   Analysis of the overall testing results would help us optimize the detection and response system in future testing.

- ***Response Tier:*** This is a response layer, which is very useful and is the retaliation point of our entire system. This layer includes our ZigBee Controller (ZC) modules, drones, and the coordinates of the sensors from which we receive signals will be provided to the drones so that the drones perform targeted spray on the insects. If the insect had gone somewhere else by the time drone arrived, then we could increase the target region by including its neighborhood areas. In Figure 5, we have shown the whole

framework, and apparently, four different drones are visible, but these will be only one or two drones, and they will continue to spray on the signals of different sensors on their specific routes.

- *Farmer Tier:* We propose a web interface for the farmer to view daily activities, including daily reports showing how many sensors sent detection packets and how many times drones sprayed on the crop. Figure 6 shows the full flow of the system, how the system receives signals and how it responds to them.

*3.3. Smart Response Mechanism*

In this section, we have presented a smart response system, including an algorithm for predicting how to automate the operations of drones.

The simulation field is divided into clusters uniformly, and IoT nodes within each cluster are also placed uniformly. The transmission power of all the sensors in the field is kept the same while the range of the gateway node is high so that it can respond to all the sensors simultaneously and send detection signals to the drones.

In Figure 8, those clusters are visible, with several sensors placed in each cluster. Second, the base station has a list of coordinates of each cluster, and the base station which receives the message from any field sensor knows the location of the sensor and its cluster from where the message came.

We have also developed coordinate-based low-altitude-flying flight paths so that the drone can reach the cluster in the shortest possible time after getting the coordinates-based message from the base station. We also know that flying moths do not move very fast. The speed of a drone is much faster than the flight of Moths.

Drones and base stations will also have two types of communication. The first is that the drone is in the hangar, and the base station will send a message to it to go into the cluster and spray. Otherwise, the drone will be flying in the field, and the base station will send another message during the flight to another cluster. The base station also can create a full schedule and send drones to the field simultaneously so that drones can spray on different clusters in a single flight. The drone will remain in constant contact with the base station even after taking off from its hangar.

*3.4. Algorithm Operations and Analysis*

Our Drone movement prediction algorithm consists of three parts, (See Appendix B at the end of the paper for the algorithm). In the first part, all the devices start their work so that they can instantly synchronize with each other. The reason for separating this portion is that when different wireless devices start their communication, it takes some time, and they send initial bits (beacons) to let each other know that we are ready for further communication.

The second part of the algorithm shows the actual operation. The sensors will first send a message to the base about the presence of insects, and then the base station will send a message containing the sensor's coordinates to the drones so that the drones can go there and respond (pesticides) accordingly. For example, if we look at the first case or the first condition of the algorithm, it is an ideal situation that the base receives a message from the sensor in the field about the presence of flying insects and the base immediately dispatches the drone to the cluster from which the message came, and it should be completely sprayed to protect crops from insects. This is the scene we have shown in the simulation in which the drone goes there and sprays after giving the message to the sensors, as shown in both parts of Figure 9.

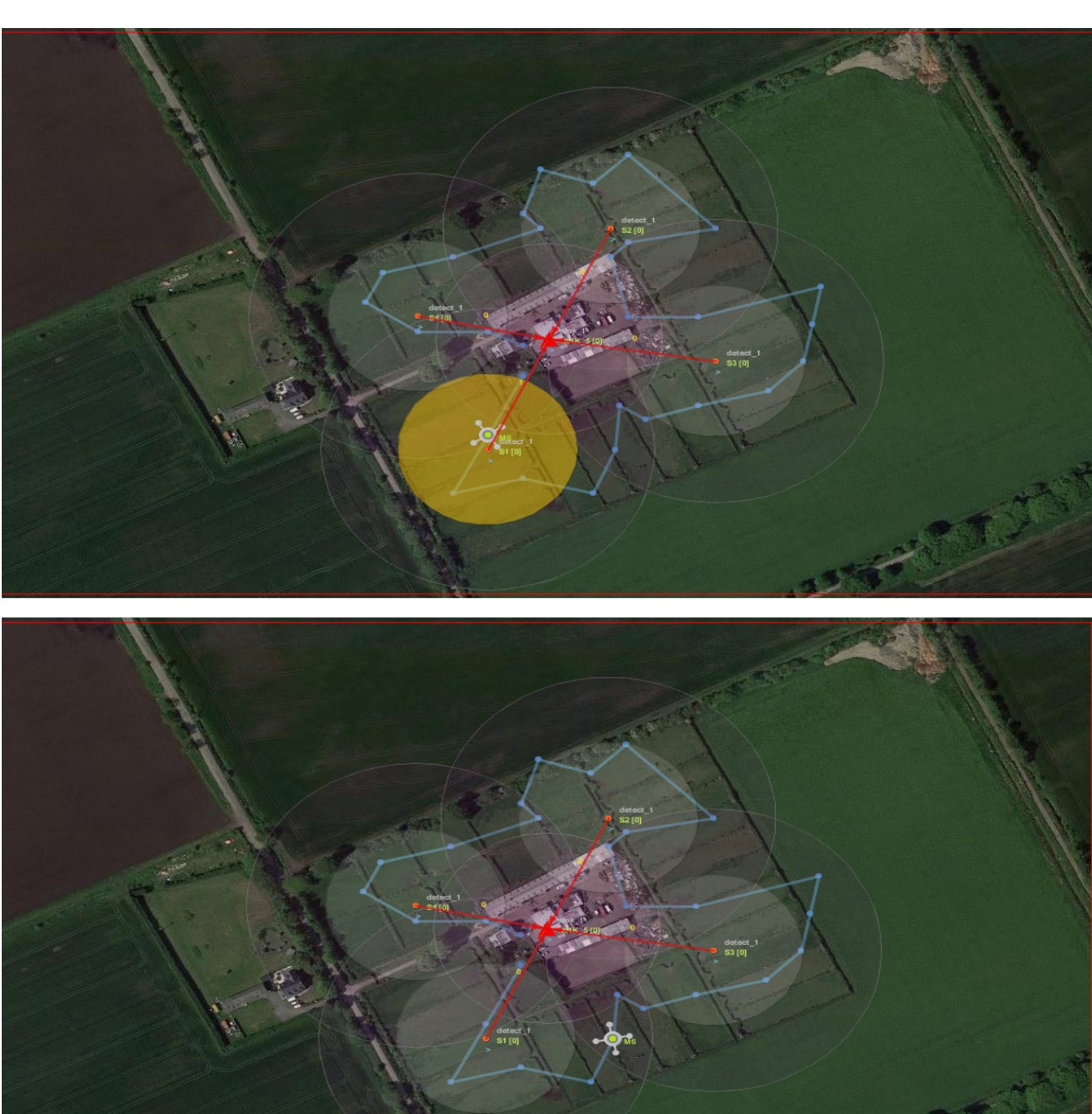

**Figure 9.** Drones and insects moving to different clusters in Field.

Once the drone is finished with spraying, it checks for any other messages from the base station about the presence of pests in any other cluster. If yes, then it will travel straight to that cluster for spraying, and then it will come back to the hanger.

In the second case, if the drone has come out of its hanger and only one message comes to the base, and no other message arrives later, the base will call back the drone, and it will return without spraying. In this case, it is more likely that the sensor sent the message for reasons other than the presence of pests.

If the base station receives messages from many clusters at the same time, the base station will make the drone a complete path to decide which cluster to spray first and then which other cluster to spray later. So that complete spraying can be completed at the same time, and if more than one drone is needed for this work, they can also be used. In addition, if there are still permanent messages coming from any cluster, the base station

will send a camera drone to see why the messages are coming from there. Maybe the sensor is generating messages because of something other than the presence of bugs.

Suppose the above three scenarios have to be implemented accurately so that the crop can be protected in any case. Similarly, all possible scenarios will be applied to the crop, and the algorithm will be applied to the framework so that in case of any scenario, the crop will be safe, the quality of the crop will be maintained, and the farmer will be saved. The second part of the algorithm is very important, and even after this part of the algorithm is properly run and implemented.

If a problem still exists, then the third part of our algorithm will solve the remaining problems. For example, If a sensor is sending a message despite the drone being sprayed and monitored by a camera, then as a last step, we will send a human helper there to monitor the problem. If there is a problem that can be seen, then they will report to the control room immediately, and immediate action can be taken.

We know that pesticide sprays affect both the quality and quantity of the crop, and there is a separate cost to the farmer for a pesticide which reduces its profit. This whole framework is an effort to maintain the quality of the crop as well as save the farmer money and increase profit. We know that this is a distributed artificial intelligence system and its many factors work together. All the sensors and devices working together in the right way is very useful for the framework and provides very good results.

One of the hurdles in adopting technological solutions in developing countries is cost. We now present the cost estimation at an abstract level because it is inevitable that additional money will be required to implement it by either farmers at the individual level or by the government. To install the proposed solution in the agriculture field, including the cost of installing sensors and base stations with gateways and controllers. The drones that are part of our response team will require paying a significant price. We also have to pay for monitoring them, which was not included in our expenses earlier. This is all the extra money the farmer will have to pay while only the cost of medicines/pesticides will be saved, which is not much more than that. Due to these additional costs, farmers are reluctant to adopt any new technology. In addition, farmers will have to learn how to use it, which will be a difficult task for them.

However, with regards to the quality of the crop, it will be much better because the spray of the medicine greatly reduces the quantity and quality of the crop. Adopting the technology will require a one-time investment, but the running cost is low, and these devices could be used in many more crop seasons. If we estimate how much our system will cost, it will cost a few thousand dollars in the beginning, but it will be useful for a long time, and if you learn to use these things well, you're running cost will be very low.

The cost of this system may be high, but with the price, we also consider it necessary to compare the response time of our framework with the old method that the farmers themselves used to perform all this work. Definitely, this framework will work 24 h a day where farmers can only work for a limited time while its response time will be much less than a human response. While the response of this system can be further improved, human capabilities can only work to a limited extent.

Insects can be detected quickly through this framework. Second, they can be sprayed quickly and automatically through an automated system. In this regard, limited medicine will have to be sprayed, and the cost of medicine can be saved. At the same time, the quality of the crop can be increased, after which the profits of the farmers can be increased.

## 4. Simulations

We have also conducted simulations so that we can be more prepared when putting our concept into practice. For this, we have used IoT based cup carbon simulator. In the simulation, the drones fly at their specific routes showing their proper and quick response. The above prediction algorithm can also be applied here.

We have implemented an ideal scenario in which the drone goes to the location of the sensor and sprays in response to the sensor's message. The rest of the scenes are yet to be implemented, which are part of our future work.

### 4.1. Simulation Setup

For this, a state-of-the-art Cup Carbon [40] IoT 5.0 simulator is used. Cup Carbon is a Smart City and Internet of things wireless sensor network (SCI-WSN) simulator. We have created a whole field in which a flying Cotton insect (Moth) is roaming around different plants, and wherever the Pest goes, the IoT sensors of that cluster send messages to the base station through the LoRa communication module. In response, the base station sends the UAV (Drone) to the affected cluster, and the drone goes there and sprays the pesticides on the affected plant. For this purpose, different clusters have been formed in the cup carbon so that a flying insect can be seen moving around in different clusters similar to a real scene, as shown in Figure 8.

IoT sensors are mounted on the edges of each cluster so that as soon as a worm tries to cross a wall made from beams generated by sensors, the sensor sends a signal to the nearest base station. (In the real framework, we have used sharp sensing devices as IoT sensors which are lightweight, low-cost sensors suitable for small movement detection).

### 4.2. Assumption and Operation

We supposed that the simulated UAV was based on the AGT-15 Drone sprayer, which is considered an advanced spraying drone for agriculture, which is a hexacopter with a Multi liter spray tank.

We have just shown in our simulation that:

- Each cluster has sensors connected to the nearest base station via the LoRa Communication module.
- The whole proposed system will currently have one drone/UAV that is connected to the base station, and as soon as a message from a sensor arrives, it goes there and sprays according to the coordinates given by the base. (However, we have proposed multiple drone operations in our predictive Algorithm. An advanced camera-equipped drone has also been proposed, which will be part of our system in the future.)
- In our simulation, some clusters and base stations are connected by a single hop, while some are connected by double or multiple hops.
- In this simulation, in some clusters, we operated the drones on the base's signals, while in some places, we tried to operate the drones manually.
- Everything in the simulation is pre-determined; for example, how long will the simulation last? How long will it take for the drone to reach the sensor (if it is reached automatically by the sensor's messages), how long will it take for us to arrive there (if we operate UAV manually), or how long will it take for the worm to circulate in the entire field?
- Drones that arrive on a single hop will certainly spend less time to reach, but drones that access a sensor or worm on a double hop will have a longer time. (The same thing will happen in the real scene drone that arrives from a distance will have a longer arrival time, while a nearby drone will quickly reach its target.)
- According to the design topology, following the sensor signals, the drone will move into specific clusters and spray on insects or infected plants. Figures 9 and 10 show drones and worms hovering in different clusters of the field, and when the worm enters a cluster, the simulator points it out, as seen in Figure 10.
- The simulator has limitations, so now we can only deliver drones to clusters from which the message of insect identification has come. We are not yet able to chase insects or send drones to a specific place where the worm is flying at the moment.
- We have also created a specific path for the flying moth in the simulation through which the moth/insect passes, and the clusters it passes through generate signals of its presence, and accordingly, the drone follows it. The simulator identifies the

worms and the presence of the drone, and when the worm enters, a cluster simulator highlights this particular cluster so that both the moth and the drone operation can be seen.

- After the spray operation, the drones return to their designated hangars and then wait for the next signals.
- If the field is large, then more drones will have to be used, but if the field is small, then only a small number of sensors and drones can work. Nowadays, there are a lot of companies making spray drones, so their price has also come down a lot.
- Our drones do not yet have cameras, but in the future, if we equip our spray drones with cameras, we will be able to see the insects and chase them and destroy them on the spot.
- One of the problems with our system may be that the signals are from an insect that is not harmful, and we have to eliminate it with our drones or spurs. This problem can also be solved when we put cameras on our drones and see if it is a harmful worm or not.
- But if the Moth/Worm is not found there, how will the drone chase it or return after spraying the plants? This is a question to which we are looking for an answer.

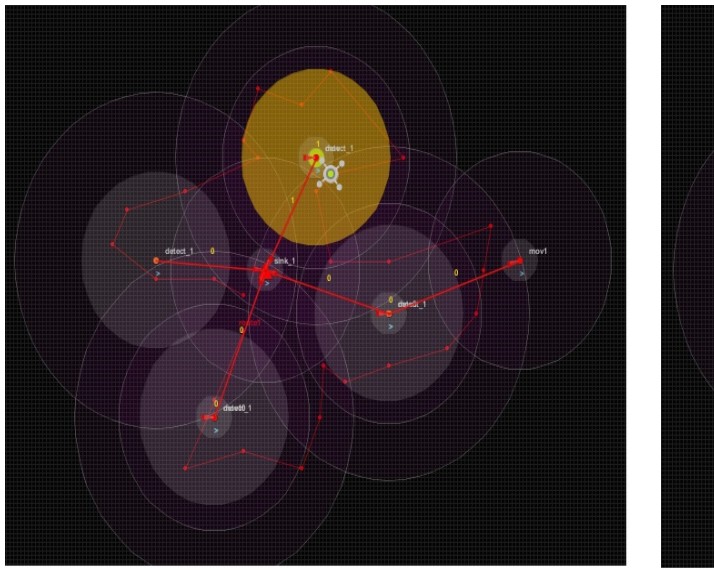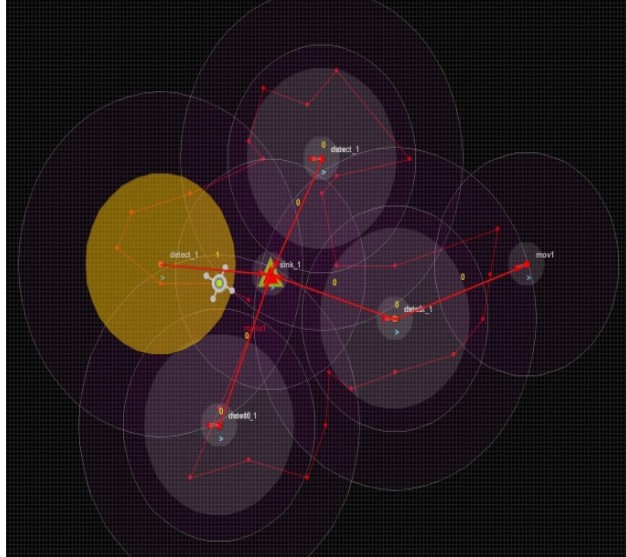

**Figure 10.** Drone chasing flying moths in different clusters.

### 4.3. Signal Patterning during Insect Detection

Besides detection, we are not yet examining the signal patterns here to see which insects are producing which signals, and for this, we will have to change our sensors and add signal processing which is the goal of our future work. In our previous work on this pest detection project, we tested what signal a moth can produce, and if the size is large, it produces a solid and large amplitude signal. Otherwise, it produces a small amplitude signal which is also weak.

### 4.4. Statistics Analysis

During the simulation, we performed a lot of tests. There was some single-hop communication, and some mobile sensors were involved in multi-hop communication. There was nothing more different in single and multi-hop communication, just a few milliseconds delay in later type of communication. You can see in Table 2 the difference between a single hop's mean value and a multi hop's mean value. We tried to review it and had to take a statistical analysis for it, which proved to be beneficial, and we found that this difference was significant. We ran simulations for this test several times and used 26 samples to attain good and stable results.

**Table 2.** Group Statistics.

|  | Hop Types | N | Mean | Std. Deviation | Std. Error Mean |
|---|---|---|---|---|---|
| Time Consumed in Milliseconds | Single Hop | 26 | 0.000008240 | 0.0000000000 | 0.0000000000 |
|  | Multi Hop | 26 | 0.003017301 | 0.0006120497 | 0.0001200328 |

As seen in Table 3, When we performed an independent sample *t*-test on it via SPSS Package, it seems that the milliseconds are the difference when the hop is increased, the *t*-test proved that it is a significant difference that will not be accepted in our communication scenario because we need a rapid response to catch pests and as if we go more to hops, this difference will increase further.

**Table 3.** *t*-Test Results and Statistical Analysis of Single and Multi-Hop.

| | | Independent Samples Test | | | | | | | |
|---|---|---|---|---|---|---|---|---|---|
| | | Levene's Test for Equality of Variances | | *t*-Test for Equality of Means | | | | | |
| | | | | | | | | 95% Confidence Interval of the Difference | |
| | | F | Sig. | t | df | Sig. (2-Tailed) | Mean Difference | Std. Error Difference | Lower | Upper |
| Time Consumed in Milliseconds | Equal variances assumed | 4.340 | 0.042 | −25.069 | 50 | 0.000 | −0.0030090606 | 0.0001200328 | −0.0032501536 | −0.0027679675 |
| | Equal variances not assumed | | | −25.069 | 25.000 | 0.000 | −0.0030090606 | 0.0001200328 | −0.0032562728 | −0.0027618483 |

## 5. Conclusions and Future Work

In this paper, we have proposed early detection and response system for cotton pests, which consists of IoT technology sensors and drones. We have proposed a complete framework, a continuation of our old work. In terms of cost, we know it is an additional burden on farmers, but in terms of crop quality, the deal is not particularly expensive. We did not implement this proposed system, but we have put sensors in the field and seen the results, which were very appropriate. In this paper, we have tried to show our framework through simulation.

After this work, many challenges are waiting for the attention of our sensor industry, researchers, and academics. The challenges we see are as follows.

- Availability of required IoT sensors to identify different cotton pests. (Although some sensors are available in the market as we have used a sensor in our work. However, there is still room for more smart sensors to be designed with better results).
- Getting real insect detection signals through IoT sensors and further refining them so that no other technology is needed is also a challenge. This is the challenge of the sensor design market.
- Making all these sensors and drones on a commercial basis is no longer considered a challenge. So that their prices can be reduced and farmers can implement such technology in their fields
- Just as we have worked on identifying pests of one crop field, expanding one system or framework and using it to identify other pests is also a challenge, and we are working on it further.
- Because IoT sensors are too small, meeting their energy needs is also a challenge so that they can last a whole season without any extra charge or battery, and they also need to be protected from large animals and protected from theft.

Along with the challenges, there are a few of our future tasks that, if completed, could make this framework even better and more streamlined in terms of results, which are as follows:

- One of our tasks in the future is to expand the framework for pest detection of other crops as well, and the framework that we have just developed for cotton pests is based on pests of other crops. It can also be applied in the same way as it has been applied to cotton bollworms.
- The usefulness of this framework can also be enhanced by creating a web interface and linking to the cloud.
- From this framework, we are currently only detecting pests, but if we also use weather and environment monitoring sensors in our framework to detect the arrival of pests due to climate change, then our framework can be more effective.

Our framework will truly help modernize farming. However, some technical and commercial issues remain unresolved and need to be addressed. We will make this predictive algorithm smarter in the future and design a fully automated system that can run without any human help and protect crops from pests.

**Author Contributions:** Conceptualization, S.A. and A.N.; methodology, S.A., A.N. and A.M.; software, A.M. and S.A.; validation, K.A. and H.A.; formal analysis, A.N. and S.S.A.; investigation, S.A. and A.M.; writing—original draft preparation, S.A., A.N. and A.M.; writing—review and editing, K.A., H.A. and S.S.A.; visualization, A.M.; supervision, A.N. and K.A.; project administration, A.N.; funding acquisition, A.N., H.A. and S.S.A.; All authors have read and agreed to the published version of the manuscript.

**Funding:** This research is funded by the Deanship of Scientific Research, Islamic University of Madinah, Madinah, Saudi Arabia.

**Institutional Review Board Statement:** Not applicable.

**Informed Consent Statement:** Not applicable.

**Data Availability Statement:** Not applicable.

**Acknowledgments:** We would like to acknowledge the funding support from the Deanship of Scientific Research, Islamic University of Madinah, Madinah, Saudi Arabia, and we would also like to acknowledge the support from Sindh Agriculture University Pakistan in experimentation.

**Conflicts of Interest:** The authors declare no conflict of interest.

## Appendix A

---

**Algorithm A1:** Detection algorithm.

---

1. *Initialize serial and Wireless communication*
2. *Sensor Value = Analog Read (Sensor Pin) (Get value from all sensors separately, every sensor represents by a separate pin)*
3. *Distance = Sensor Value*
4. *For (sensor i = 1 to n; do)*
5. *If (distance > 1 && distance < 4) received at any of the analog Pin*
6. *Mark Detection = true*
7. *Then print on Serial "BEE DETECTED" or "Bee Detected on Sensor # 1"*
8. *Send a message to the gateway node.*
9. *Delay (from 10 ms to 1000 ms) (It is denoted by the sensing frequency of the sensor, how much time the sensor will take for a new sample or data)*
10. *The gateway sends a message to a drone as per the received coordinate.*
11. *End*

**(In a detection algorithm, important parameters like sensitivity and delay could be fixed according to precise body dimensions and the movement of target species.)**

---

## Appendix B

---

**Algorithm A2:** Proposed drone movement predictive algorithm.

---

1. Initialize serial and wireless communication Sens (field);   // *All sensors in the field*
2. (sensor = S – Cluster = C) Base (Control);   // *Base station in Control Room (Base = B)*
3. Drone (Hang);                                //*Drone in the hanger near*
4. *Control* Station (Drone = D)
5. Packets = P                                  // *Message from Sensor to Base and Base to Drone*
6. Hanger = H                                   // *Drone Station*
7. Farmer = F                                   // *Monitoring Assistance*
8. **Camera Drone = D-Cam**                     // *Drone Equipped with Camera Start beacons*
   **Initialize serial and wireless communication Sens (field);**
9. **while loop Starts**
10. **if**
11. B get P from S(n)                           // time T1
12. Then send D to S == C                       //for Spray at T2
    a.  *Spray pesticides;*
13.       **for** (int i=n; i<= S(max)-n; i++)
14.         **if**
15.           B get P from S (i) == C (i)
16.           Then send D to S (i) == C (i)
17.         *Spray pesticides;*
18.       **for Ends**
19.       **if** B not-get P from S (n)                // False Alarm
20.         D moved to Hanger
21.       **End if**
22. **else if**
23.       B get P from S (2) == S (3) == S (1) == S (n)    // Multiple Insects in Field
24.       B send M to D through GSM
25.       Send D to S (2) == S (3) == S (1) == S (n)       // Multiple Clusters
26.       *Spray pesticides;*
27. **else if**
28.       B get P from Multiple S(n)                    // Continuously received False Alarms
29.       Then send P to D-Cam == F to S(n)
30.       *Perform manual Monitoring through Camera Drone;*
31. // Continuous Alarm / Due to some other object
32.       **if** B get P from S (n) continuously
33.         B send M to Farmer                          // to visit personally in field
34.         *Perform Monitoring through Human assistant;*
35.       **End if**
36. **End if**
37. **while loop Ends**
38. **Deactivate serial and wireless communication Sens (field);**

---

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
