# Peer review of "IoT-Based Cotton Plant Pest Detection and Smart-Response System"

_applsci, doi:10.3390/app13031851_

Round 1

Reviewer 1 Report

 IoT- based Cotton Plant Pest Detection and Smart-Response System this manuscript presents a framework that detect insects through motion detection sensors and then get a smart automatic response using drones based targeted spray. The contribution sounds good

The author should take these comments into consideration for improve the quality of the paper

·     In this research, we have proposed a complete system, relying on IoT, which….”” This paragraph should be modified.

·     The organized of the paper should be rewritten.

·     The Figure 2 and Figure 3, are listed without caption in the paragraphs.  

·     Proposed Drone Movement Predictive Algorithm should be removed into appendix.

·     Figure 5 needs to be draw and to be modified.

·     An extensive English improvement need for many paragraphs  such as “Now we want to complete our previous task [38] and design a complete system in 287 which our drones will immediately respond to the signals of the sensors in the field and 288 the crop will be protected from pests and diseases.”

·     The authors should be discussed what are the criteria that have been performed to cluster the IoT nodes.

·     The way of discussing or call out the algorithm need to be improved such as “As shown in following algorithm. We suggested this algorithm for similar task in our 314 previous paper [38]. It may be applicable here as well” A pseudocode or the main steps or add it in the appendix. Also any algorithm should have a label.

·     Any flow chart should be added to present all methodologies for this paper.

·      Figure should have the same style in all paragraphs i.e we see sometimes small letter and sometimes capital letter

Reviewer 2 Report

Reviewers thank the authors for the submission of the manuscript. In this paper, authors consider the issue of smart pest detection and management of cotton plants and propose an IoT framework first to detect insects through motion detection sensors and then get a smart automatic response using drones based targeted spray. In this proposed method, they have also explored an interesting combination of ZigBee communication technology and drones to improve field surveillance and then proposed a predictive algorithm for a detection response system using a decision-making theory.

In order to further improve the quality of the manuscript, the following amendments are proposed:

  1.  The novelty of the presented study is not clearly indicated. Please add a short description, in the pre-last paragraph of the Introduction Section, of the innovative characteristics introduced by your team regarding the IoT sensor network and information exchange, along with data management and analysis procedure.
  2. There is no clear comparison with the state-of-the-art approaches. Also, authors can include the following recent publications in the literature survey and follow the writing style of papers. 

·        Bhanu, K. N., Jasmine, H. J., & Mahadevaswamy, H. S. (2020, June). Machine learning implementation in IoT based intelligent system for agriculture. In 2020 International Conference for Emerging Technology (INCET) (pp. 1-5). IEEE.

·        Sharma, R. P., Ramesh, D., & Edla, D. R. (2022). IoFT-FIS: Internet of farm things based prediction for crop pest infestation using optimized fuzzy inference system. Internet of Things, 100658.

·        Sahana, K. (2021, August). Farm Vigilance: Smart IoT System for Farmland Monitoring and Animal Intrusion Detection using Neural Network. In 2021 Asian Conference on Innovation in Technology (ASIANCON) (pp. 1-6). IEEE.

·        Sharma, R. P., Ramesh, D., Pal, P., Tripathi, S., & Kumar, C. (2021). IoT-enabled IEEE 802.15. 4 WSN monitoring infrastructure-driven fuzzy-logic-based crop pest prediction. IEEE Internet of Things Journal, 9(4), 3037-3045.

·        Pal, P., Sharma, R. P., Tripathi, S., Kumar, C., & Ramesh, D. (2021). Genetic algorithm optimized node deployment in IEEE 802.15. 4 potato and wheat crop monitoring infrastructure. Scientific Reports, 11(1), 1-12.

·        Bhanu, K. N., Mahadevaswamy, H. S., & Jasmine, H. J. (2020, July). Iot based smart system for enhanced irrigation in agriculture. In 2020 International Conference on Electronics and Sustainable Communication Systems (ICESC) (pp. 760-765). IEEE.

  1. The infrastructure seems disconnected from the rest of the text and very poor. How was it used? How many days was its operation, and what kind of data was produced? It is stated in the manuscript the drone is used to analyze the data; however, it is not clearly mentioned about the programming platform as well as the data analysis scheme. Could it be adopted in nearly real-time application scenarios?
  2. Authors have also been suggested to make clear the number of drones taken into consideration. In Assumption and Operation, it is mentioned each cluster has its own drone; however, the smart response mechanism section states drones can go to a different cluster which makes it confusing why such assumptions are being made.

  1. Please thoroughly check the entire manuscript to proofread it since typos and syntax errors are met. Also, please keep consistency in the tense of verbs used. A few of the sentences are confusing and hard to follow. For example, In the related work section, the first paragraph line “various fields of artificial intelligence and sensors and flying drones. The following works are similar to our work but some of the works are not similar to our work but the concept of our work is definitely there whereas the technology is somewhat different.” Again, in line “In [17], a forest fire is a natural disaster that spreads very quickly and burns everything in its path” the sentence is grammatically incorrect. The line “It will cost about nine hundred dollars.” seems incomplete.
